# Transmission-Blocking Vaccines against Schistosomiasis Japonica

**DOI:** 10.3390/ijms25031707

**Published:** 2024-01-30

**Authors:** Chika P. Zumuk, Malcolm K. Jones, Severine Navarro, Darren J. Gray, Hong You

**Affiliations:** 1Infection and Inflammation Program, QIMR Berghofer Medical Research Institute, Herston, QLD 4006, Australia; chika.priscazumuk@qimrberghofer.edu.au (C.P.Z.); m.jones@uq.edu.au (M.K.J.); severine.navarro@qimrberghofer.edu.au (S.N.); 2Faculty of Medicine, The University of Queensland, Herston, QLD 4006, Australia; 3School of Veterinary Science, The University of Queensland, Gatton, QLD 4343, Australia; 4Centre for Childhood Nutrition Research, Faculty of Health, Queensland University of Technology, Brisbane, QLD 4000, Australia; 5Population Health Program, QIMR Berghofer Medical Research Institute, Herston, QLD 4006, Australia; darren.gray@qimrberghofer.edu.au

**Keywords:** *Schistosoma japonicum*, zoonotic schistosomiasis, transmission blocking vaccines, vaccines, water buffalo

## Abstract

Control of schistosomiasis japonica, endemic in Asia, including the Philippines, China, and Indonesia, is extremely challenging. *Schistosoma japonicum* is a highly pathogenic helminth parasite, with disease arising predominantly from an immune reaction to entrapped parasite eggs in tissues. Females of this species can generate 1000–2200 eggs per day, which is about 3- to 15-fold greater than the egg output of other schistosome species. Bovines (water buffalo and cattle) are the predominant definitive hosts and are estimated to generate up to 90% of parasite eggs released into the environment in rural endemic areas where these hosts and humans are present. Here, we highlight the necessity of developing veterinary transmission-blocking vaccines for bovines to better control the disease and review potential vaccine candidates. We also point out that the approach to producing efficacious transmission-blocking animal-based vaccines before moving on to human vaccines is crucial. This will result in effective and feasible public health outcomes in agreement with the One Health concept to achieve optimum health for people, animals, and the environment. Indeed, incorporating a veterinary-based transmission vaccine, coupled with interventions such as human mass drug administration, improved sanitation and hygiene, health education, and snail control, would be invaluable to eliminating zoonotic schistosomiasis.

## 1. Introduction

Schistosomiasis, caused by human trematode blood flukes, is globally ranked as one of the most devastating neglected tropical diseases (NTDs), prevalent in poor and developing nations in Asia, Africa, the Middle East, South America, and the Caribbean [1,2,3]. The most prominent species affecting humans (Appendix A) are *Schistosoma haematobium*, the agent of urogenital schistosomiasis, and *S. mansoni* and *S. japonicum*, both causing hepato-intestinal schistosomiasis [4]. Co-infections with other diseases such as malaria, tuberculosis, human immunodeficiency virus (HIV), hepatitis B virus (HBV), and hepatitis C virus (HCV) in regions with endemic schistosomiasis result in increased morbidity and mortality in humans [5,6,7,8,9,10,11].

Schistosomiasis has an estimated disability-adjusted life year (DALY) of 2.5 million, according to the neglected tropical diseases road map, approved by the World Health Assembly [1,2]. The disease is endemic in 78 countries, with at least 251 million people requiring praziquantel (PZQ) preventative treatment, the only therapeutic regimen for schistosomiasis supplied by the Merck Schistosomiasis Elimination Program [2,12]. Praziquantel is effective and cheap; however, it is limited by various factors, including a short half-life in humans (lasting just 3 h), poor efficacy against the larval schistosomulum stage of the parasite, the inability to prevent reinfection, and the potential risk of drug resistance [4,13,14,15], particularly as the drug is used for the treatment of veterinary schistosomiasis and other helminth infections. While preventive measures such as improved sanitation, snail mollusciding, and mass drug administrations (MDAs) have been effective in reducing the transmission of this disease, the development of transmission-blocking vaccines holds great promise for further controlling its spread. However, progress has been slow in this area [16]. There are currently three vaccines against *S. mansoni* infection in various stages of clinical trials. These vaccines are *S. mansoni* tegument tetraspanin Sm_TSP-2, formulated on Alhydrogel with Glucopyranosyl lipid A (GLA-AF), *S. mansoni* 14 kDa fatty acid-binding protein (Sm 14), and *S. mansoni* calpain (Smp80), both formulated in GLA-Se (GLA in stable oil emulsion) [17,18,19,20] and *S. haematobium*, Sh28GST, adsorbed to Alhydrogel [21].

*Schistosoma japonicum* causes a zoonosis endemic in Asia, including the Philippines, China, and Indonesia. It can infect over 40 species of wild and domestic animals (including dogs, pigs, rats, goats, sheep, and bovines) [3,22,23,24]. The broad host range of this species and its transmission through freshwater snails significantly increase the complexity of control strategies and place this species firmly as a One Health concern. The parasite affects humans not only directly through human disease but also places a huge burden on communities economically since schistosomiasis incapacitates the productivity of work animals [3]. However, various field studies have identified that bovines, particularly water buffalo (*Bubalus bubalis*), are responsible for up to 90% of environmental egg contamination as major animal reservoir hosts in the transmission of *S. japonicum* to humans in China, Indonesia, and the Philippines [24,25,26,27,28,29,30,31,32,33].

## 2. Life Cycle and Immunopathology

The life cycles and distribution of schistosomes depend on the availability of snail intermediate hosts. Unlike the mammal-parasitic stages, intramolluscan stages are highly host-specific. *S. japonicum* infects snails of the genus *Oncomelania*, *S. mansoni* infects snails of the genus *Biomphalaria*, and *S. haematobium* infects snails of the genus *Bulinus.* Schistosome eggs, released with human or animal faeces (*S. mansoni* and *S. japonicum*) or urine (*S. haematobium*) into the environment, hatch after exposure to fresh water, liberating the larval stage, the miracidium, which seeks and penetrates a susceptible snail (Figure 1). In snails, substantial asexual reproduction takes place across at least two generations [34]. Thus, the miracidium transforms into a stage termed the mother sporocysts, which give rise to daughter sporocysts, which in turn give rise to the next water-transmissive stage, the cercaria. Cercariae escapes the snail to penetrate the skin of humans/animals through contact with contaminated water. Within the skin and sub-dermis, the cercaria transforms into a host-adapted schistosomulum, a parasitic larval stage. This larva then enters the circulation, travelling to the lung and eventually through the portal circulation to the liver, where male and female worms develop and pair [35,36]. Worm pairing is necessary for developing the ovary and vitelline glands in female worms, leading to embryogenesis and egg production [37,38,39,40]. The paired adults migrate to the mesenteric veins of the intestine (for *S. mansoni* and *S. japonicum* infections) or venous plexus of the bladder (*S. haematobium*), where eggs are shed in stool or urine at 4–6 weeks post-infection. About one-third to one-half of these do not reach the external environment and are responsible for *Schistosoma* pathogenesis.

Adult *S. japonicum* females, characterized by a powerful fecundity system, can produce a 3–15 fold higher egg output (about 1000–2200 eggs per day) [36,41]. With around half of the eggs trapped in host tissues, they form larger aggregates compared with other schistosome species (including *S. mansoni* and *S. haematobium*) [35,36,41,42,43]. Chronic infection occurs due to the progressive accumulation of parasite eggs within host tissues, including the intestines, liver, and, in some rare cases, the gall bladder [44]. The trapped eggs release soluble egg antigens consisting of proteolytic enzymes and antigenic glycoproteins. These antigens provoke inflammatory immune reactions, leading to the migration and assembly of inflammatory cells. Consequently, this process results in the formation of granulomas and subsequent fibrosis, aiming to encapsulate the eggs and neutralize their toxic antigens [34,45]. Macroscopically, the formation of granulomas and fibrosis manifests as colonic polyposis and peri-portal fibrosis (for hepatic-intestinal schistosomiasis) and can also cause deterioration of the liver, the development of ascites, jaundice, and resulting complications in the lung and central nervous system, which can be fatal if untreated [46,47,48].

Most of our knowledge of the immunopathology of schistosomiasis arises from murine studies and has been reviewed extensively [34,36,42,46,49,50,51,52,53,54,55,56,57,58,59]. An initial CD4^+^ T helper (Th) 1 immune response is generated against the migratory schistosomula, leading to the secretion of pro-inflammatory cytokines such as interferon (IFN)-γ, IL-1, IL-6, IL-12, and alpha Tumour Necrosis Factor (TNF-α) [42]. Following egg deposition, at approximately six to eight weeks after infection with *S. japonicum*, CD4^+^-Th2 cells are activated [60]. IL-4, IL-5, and IL-13 are then released, recruiting pro-inflammatory monocytes, neutrophils, eosinophils, hepatic stellate cells (in the liver), and macrophages around the eggs to form granulomas, collagen deposition, and fibrosis, leading to the destruction of the eggs [34,50,51,52,53,54]. Th2 cells also stimulate B cells to produce immunoglobulins, such as IgE, which have been associated with resistance to reinfection after praziquantel treatment [34,61]. Regulatory T cells (Tregs) and B cells (Bregs) stimulate the production of IL-10 and TGF-β and other anti-inflammatory cytokines that downregulate granulomatous reactions in chronic infections, and inhibiting them has the potential to enhance vaccine efficacy [34,50,53,55,56,62,63,64,65,66]. Furthermore, Th17 cells with IL-17, Th9, and follicular T helper cells are also important for hepatic granuloma formation and the regulation of the immune response to *S. japonicum* infections [56,57,58,59]. There have been limited studies on schistosomiasis immunopathology in the natural reservoir host for schistosomiasis japonica. It is worth noting that a study on the immune response of water buffalo to schistosomiasis found a strong Th2 response during cercarial skin penetration and a moderate type-1 response when the larvae migrate to the lungs [67]. Another study suggested that older water buffalo can develop natural resistance to and self-cure (sharp decrease in worm burden due to parasite clearance by both immune and non-immune factors) from schistosomiasis, showing increased levels in IL-4, IL-10, IFN-γ, and specific IgG antibodies at 4–9 weeks post-secondary infection [68,69]. However, further investigations are needed to better understand the immunological mechanism in bovines, which could have important implications for developing new vaccine targets and control strategies, particularly in the deployment of vaccines to younger animals since older animals self-cure [69].

## 3. Rationale and Implications for the Development of a Veterinary-Based Transmission-Blocking Vaccine

Effective control of schistosomiasis involves reducing the release of eggs from infected primary hosts into waterways, which subsequently infect the snail intermediate host [29,70]. Repeated exposure of humans to contaminated fresh water containing infected snails can lead to skin penetration by cercariae, causing infection. To effectively control schistosomiasis, it is crucial to identify the main reservoir hosts of *S. japonicum*, even though the parasite has numerous hosts. Studies have shown that water buffalo are the most significant reservoir host in Asia, and controlling the disease in this host should be the main objective for disease elimination [28,69]. Bovines, especially the water buffalo, play important roles in schistosomiasis transmission, especially in the Philippines, where the prevalence of schistosomiasis in water buffalo is as high as 95% [31,71,72,73,74], while in China, the prevalence in water buffalo ranges from 8 to 27% [24,27,29,32,75]. Field investigations performed in Indonesia also indicated that buffalo and cattle were major sources of environmental egg contamination, contributing about 59% and 22%, respectively [33]. The prevalence of *S. japonicum* infection in China decreased considerably following concerted efforts to effectively control bovine schistosomiasis in endemic areas through its Chinese national schistosomiasis control program [76]. Several strategies were implemented, such as snail elimination, morbidity control through mass drug administrations (MDAs), interrupting transmission to snails by reducing egg contamination through bovine chemotherapy, and the replacement of cattle with mechanized farming in endemic areas [77,78]. As a result, only about 30,000 humans were infected with schistosomiasis in 2019, compared to the 726,000 human cases reported in 2004 [76,79]. It is interesting to note the variation in prevalence rates of schistosomiasis in different host animals in the Philippines. The prevalence rate of *S. japonicum* in water buffalo in the Philippines was previously reported to be low or even absent, and it was thought that rodents and canines were the main reservoir hosts for schistosomiasis transmission, having higher prevalence rates than water buffalo [31,80,81,82,83]. In fact, the prevalence rates of schistosomiasis in rodents were reported to be as high as 73%, while dogs had up to 19% [80,81,84]. These findings indicate that the prevalence rates of schistosomiasis can vary even in endemic areas, as highlighted in an extensive review by Tenorio et al. [84]. It is argued that the low prevalence reported in some studies could also be a result of the poor sensitivity of the tests used for stool analysis and can be improved by adjusting the FEA-SD protocol used and utilizing more sensitive techniques such as the polymerase chain reaction (PCR) tests [72,83]. It is, however, necessary to ascertain the impact of a transmission-blocking vaccine in these regions, as livestock animals, particularly water buffalo, are allowed to graze in contaminated fields. These animals are relied upon for agricultural purposes as animal traction, especially in geographical areas where bovines cannot be replaced by mechanized farming, and also for organic fertilizers [83]. A two-year intervention from 2005 to 2007 revealed that replacing cattle with mechanized farming, among other control strategies, resulted in an extensive reduction in human schistosomiasis prevalence in two Chinese villages, from 11.3% to 0.7% in one village and from 4.0% to 0.9% in the other [85]. Bovines can therefore be said to be an important reservoir host and a source of transmission of schistosomiasis to humans and other domestic animals.

Notably, the daily faecal output of water buffalo (~25–50 kg/water buffalo) is ~100-fold higher than that released by a human individual (~250 g) [3,22,31,86], emphasizing the major contribution of infected bovines to environmental egg contamination in *Schistosoma* endemic areas both in China and the Philippines [74,75]. This provides strong evidence indicating that adequate control of egg contamination in the environment by reducing the faecal egg output from hosts (especially buffalo and cattle) is critical for effectively combating this zoonotic disease [4]. Several randomized bovine intervention trials have also substantiated the notion that developing a transmission-blocking vaccine for livestock animals, especially bovines, against human *S. japonicum* infection is both a realistic and feasible complement to existing control strategies [27,29,87].

This concept is evidenced by mathematical and computational models of schistosome transmission [88]. Da’dara in 2008 [70], further supported by Williams et al. in 2019 [29], predicted that immunization of water buffalo with a vaccine that can induce 45% faecal egg reduction, integrating with PZQ treatment, will lead to a remarkable reduction in transmission, almost approaching elimination within 10 years. The mathematical model further implies that improved vaccine efficacy (>45%) has the potential to significantly shorten the time required to reach elimination. According to the first schistosomiasis vaccine meeting of the Bill and Melinda Gates Foundation and the National Institute of Allergy and Infectious Diseases in March 2013, a veterinary-based transmission-blocking vaccine acts as a type II vaccine ‘to reduce or eliminate reinfection intensity or transmission force by interrupting female worm survival or egg production’ [89]. This forms the basis of efforts to develop a veterinary transmission-blocking vaccine for *S. japonicum* as a feasible and impactful public health goal, an outcome that is in agreement with the One Health concept to achieve optimum health for people, animals, and the environment [4,16,29,90]. Furthermore, a veterinary transmission-blocking vaccine can be combined with MDA, snail control, and water sanitation and hygiene (WASH) programs to meet the WHOs plan for eliminating schistosomiasis in their roadmap for neglected tropical diseases 2021–2030 for schistosomiasis [1,4,5,12,16,91,92].

## 4. The Role of Adjuvants in the Development of Schistosomiasis Veterinary Transmission-Blocking Vaccines

Adjuvants have been utilized to improve and complement vaccine immune responses and can act as either immunostimulants, immunomodulators, delivery systems, or a combination of different types [93]. Adjuvants should be safe, increase the immunogenicity of weak antigens, enhance the dose and duration of vaccines, and be effective for every age and individual, including the elderly and immunocompromised [94,95,96]. It is, therefore, important to select suitable adjuvants that can improve the protective immune responses of vaccine candidates with fewer adverse effects. A few adjuvants have been approved for human use, including aluminum salts, MF59 (an oil in water emulsion), AS04, AS03, AS01, CpG ODN 1018 (an oligodeoxynucleotide), and recently, with the SARS-CoV-2 mRNA vaccine, lipid nanoparticles (LNP) [95,96].

Despite the benefits of adjuvants, their mechanisms of action have not yet been fully understood. Recent advances in immunology [93,95] have shown that as immunostimulants, adjuvants can be pathogen-associated molecular patterns (PAMPs), damage-associated molecular patterns (DAMPs), or Toll-Like receptor agonists that interact with receptors on Antigen-Presenting cells (APCs). They stimulate the innate immune response by activating APCs, thereby resulting in the antigen-presenting/Pattern Recognition receptor (PRR) signaling pathway and costimulatory signals, stimulation of naïve T cells, cytokine release, and the subsequent enhancement of adaptive immune response and memory [95]. As delivery systems, adjuvants assist with antigen uptake and presentation, thereby extending the bioavailability of antigens [95].

In schistosomiasis vaccine research, alum and emulsion adjuvants, which can be oil in water, water in oil, or a combination of both, are some of the earliest forms of adjuvants used. The Sm14 vaccine for *S. mansoni* and the ShGST28 vaccine for *S. haematobium* are formulated with alum (Alhydrogel) combined with Glucopyranosyl Lipid Adjuvant (GLA) in an Aqueous Formulation (GLA-AF) [21,97,98]. Alum adjuvants have been suggested to promote antibody generation and Th2 immune response by being both an immunostimulant and delivery system [93,95]. Their mechanisms of action involve enhancing antigen uptake, slow release of adsorbed antigen, recruitment of T-cells and the inflammasome, and PRR signaling [95,96,99].

Glucopyranosyl A (GLA) is a synthetic lipopolysaccharide and a Toll-like receptor 4 (TLR4) agonist that stimulates both T helper 1 and 2 immune responses and is being investigated for use in humans [95,100]. GLA-SE and GLA-AF have been said to have low toxicity and side effects [96]. Additionally, a combination of stable oil emulsions and GLA (GLA-se) has been used for *S. mansoni* calpain and Sm14 vaccines with minimal side effects [97,101].

Emulsion adjuvants are delivery systems and can also be immunostimulants [95]. The complete (CFA) and incomplete Freund adjuvants (IFA), developed in the 1930s, were commonly used in schistosome vaccine development. Although CFA was proven to be able to induce weak Th1 and Th2 immune responses, it was abandoned due to its toxicity [102]. Sj28-GST and *S. japonicum* Fatty acid binding protein (SjFABP) recombinant vaccines have been combined with IFA in various vaccine efficacy studies [63,103]. Montanide ISAs (Incomplete Seppic adjuvants) similar to IFA are shown to induce strong cytotoxic T-lymphocyte (CTL) responses and long-term immunity with minimal side effects and are also stable and easy to inject [94]. Some vaccines that use Montanide ISA as adjuvants include *S. japonicum* insulin receptors, Calpain, and paramyosin [104,105,106].

Saponins like Quil A, which is derived from the bark of the *Quillaja saponaria* tree, have been used as veterinary adjuvants and can induce both Th1 and Th2 responses [102]. Although these are generally safe, toxicity can occur depending on the route of administration [102]. When combined with the *S. japonicum* insulin receptor, worm and fecal egg reduction were induced [104,107]. Combining aluminum hydroxide gel with saponin (AS) has been used as an adjuvant, resulting in effective protection of cattle from trivalent foot and mouth disease [108].

*S. japonicum* Heat shock protein (HSP) 70 has also been used to stimulate the innate and adaptive immune response against *Schistosoma* infection [109]. The vaccine efficacy of *S. japonicum* Triose phosphate isomerase (SjTPI) and *S. japonicum* 23 kDa membrane protein (Sj23) can be enhanced when they are conjugated to HSP70 [70,109].

Cytokines such as IL-12 are also widely used as immunomodulators and potent inducers of the Th1 response [102]. It has been utilized alongside SjGST, SjTPI, and Sj23 DNA vaccines [70,110,111]. Furthermore, a series of studies have demonstrated that antibodies targeting T regulatory lymphocytes such as Monoclonal Antibodies to CD25 (anti-CD-25mAb) and Cytotoxic T-lymphocyte-associated protein 4 (anti-CTLA-4), when administered as booster doses in immunization protocols, lead to significant improvements in vaccine efficacy [62,63,64,65,112]. Further studies are required to determine the feasibility and safety of this approach.

Liposomes are nano-adjuvants structurally similar to the cell membrane and are said to be excellent drug carriers, enhancing vaccine delivery [94,113].

It has been demonstrated recently that lipid nanoparticles (LNPs), as excellent delivery carriers for nucleic acid or protein vaccines, can induce effective cellular immune responses by attracting and activating dendritic cells (DCs) [94,114]. They consist of ionizable lipids, phospholipids, cholesterol, and polyethylene glycol-modified lipids [95]. They are delivery vesicles that prolong the availability of antigens, promote antigen uptake by APCs, and escape from endosomes [95]. LNPs have been shown to improve the efficacy of the SARS-CoV-2 and influenza virus hemagglutinin recombinant protein vaccines by stimulating T follicular helper cells and B cells through IL-6 [115]. All this evidence strongly suggests that LNPs could be extended to be used as potential adjuvants for vaccine development against *Schistosoma* and other parasitic diseases.

## 5. Current Status of Transmission-Blocking Vaccines against *S. japonicum*

To date, numerous *S. japonicum* vaccine candidates have been evaluated across a range of animals (including mice, pigs, goats, sheep, cattle, and water buffalo), as we have reviewed previously [16,90]. In bovines, only a few presented more than a 45% reduction in worm burden or faecal egg output (Table 1). Worm burden reduction has been considered the “gold standard” for anti-schistosome vaccine development; nonetheless, given the key roles of schistosome eggs in both disease pathogenesis and transmission, it is crucial and pertinent to develop transmission-blocking vaccines by inhibiting parasite fecundity and egg production/viability [116]. The significant anti-fecundity impact of the type II vaccine, defined earlier, was further addressed at a schistosomiasis vaccine meeting in 2016 [117], concluding that three levels of reduction in worm burden or egg excretion induced by the transmission-blocking vaccine targeting water buffalo/cattle in endemic areas (Southeast Asia) were defined as acceptable (>60%), target (80%), and ideal (90%). This criterion did not, however, consider the impact of a combination of the veterinary vaccine with human PZQ treatment and other control measures in decreasing schistosomiasis transmission. As previously mentioned, immunizing water buffalo with a vaccine that induces a 45% reduction in faecal egg output, in combination with PZQ treatment, can significantly reduce transmission, leading to elimination in 10 years [70,88]. Furthermore, it was agreed that the preferred product characteristics (PPC) required for veterinary vaccines with low associated costs are forward and an acceptable level of protection will effectively reduce egg shedding in buffalos, potentially reducing the transmission to humans [117]. Additionally, veterinary vaccines have less stringent safety standards compared with clinical vaccines, and a ‘Grade II reactogenicity or vaccination-associated adverse event’ was deemed an ‘acceptable’ product criterion [117].

### 5.1. Radiation-Attenuated Cercariae/Schistosomula Vaccines

Eveland and Morse were among the pioneers in investigating the potential of schistosomiasis vaccines, notably discovering a remarkable 91% reduction in worm burden in mice through intra-peritoneal injections of X-irradiated schistosomula [122]. Their study marked one of the initial efforts to explore the development of vaccines for schistosomiasis. Identifying genes and pathways responsible for protection elicited by a radiation-attenuated cercaria vaccine demonstrated a Th1 bias, an antibody-mediated response, and upregulation of hemostatic pathways that block parasite migration from the lungs in mice [123,124]. A vaccine study conducted by immunizing sheep with radiated attenuated cercariae or schistosomula of *Schistosoma mattheei* generated a significant reduction (56–78%) in worm burden [125]. While a similar study using irradiated larval *S. mansoni* cercariae/schistosomula in baboons only resulted in a 20–30% reduction in worm and egg counts, in contrast, another study found that immunizing Cynomolgus monkeys with Cobalt^60^-irradiated schistosomula produced a 74% and 80% reduction in worm numbers and tissue egg burden, respectively [126,127,128]. The dose of radiation also has to be optimal so that parasites injected into the skin can exit through draining lymph nodes to the lungs, allowing for T cell recruitment [124].

To date, irradiated cercariae or schistosomula remain the gold standard for the level of protection that can be generated against schistosomiasis; however, they remain unsafe due to the possibility of reversion and impracticability for human administration, particularly with large-scale production, dose standardization, and storage issues [16,123].

### 5.2. S. japonicum Paramyosin (Sj97)

*S. japonicum* Paramyosin (Sj97) is a 97 kDa antigen localized on the muscles of all three developmental stages of the parasite and the post-acetabular (secretory) gland of the larva and cercariae [118,129,130]. Studies have suggested that paramyosin plays various roles, including structural, muscle tension (a catch phenomenon), immune evasion, interfering with the complement cascade, binding to host immunoglobulins, and inhibiting the formation of membrane attack complexes [131,132,133].

While Sj97 is not a tegumental protein, it has shown promising results as a potential vaccine candidate. Therefore, further studies are required to determine the surface composition of schistosomes, as the tegument proteins are widely considered to be optimal targets for vaccine and drug development [134]. Previous reports showed that intraperitoneal administration of *S. japonicum* IgE-specific monoclonal antibody in naïve mice resulted in significant protection against cercarial challenge and reduction in adult worms (19–58%) [135,136]. This was supported by another study demonstrating a specific anti-Sj97 IgE antibody was highly involved in protection against reinfection [61]. Immunization of mice with rSj97 also caused a significant reduction in worm burden (62–86%) [137] by stimulating B-cell and T-cell immune responses.

Furthermore, vaccination of water buffalo with recombinant paramyosin in a field trial resulted in a significant reduction in liver egg production (49%) and in worm burden (34%), indicating the crucial role of paramyosin in worm maturation and fecundity in the natural hosts [138]. A field safety assessment performed on water buffalo vaccinated with recombinant paramyosin showed no anaphylaxis detected, accompanied by a significantly increased IgG [106]. However, the progress of studies on the vaccine has been hindered due to the difficulty of the long and complex processing for expression and purification of rSj97 required for large-scale production [139]. As a result, three vaccine/challenge trials were conducted in water buffaloes, showing worm burden reduction increased from 52% to 61% when the dosage of the rSj97 vaccine was increased from 250μg to 500μg [118]. Similar efficacy was further observed in the third trial, resulting in a 58% decrease in worm burden with elevated levels of IgG compared to control trials [118].

### 5.3. S. japonicum Triose Phosphate Isomerase (SjTPI)

*S. japonicum* Triose Phosphate isomerase (SjTPI) is an important glycolytic pathway enzyme for glucose metabolism, providing energy crucial for the parasite’s growth across all stages of its life cycle. As a result, TPI has been identified as a promising target for both anti-helminthic drugs and vaccines recommended by the World Health Organization (WHO) [140,141]. Many studies have explored the efficacy of TPI-based vaccines, showing that DNA vaccines encoding TPI alone and in conjunction with Interleukin-12 (IL-12) can reduce adult worm burden (46–48%), especially female worm burden (53–59%), and liver eggs (49–66%) in pigs [142]. Not surprisingly, the vaccines had distinctly reduced female worm numbers, which correlates with studies that showed that insulin had a similar, distinct effect on glucose uptake by female worms, expanding the existing evidence that female worms consume more glucose than males for egg production [143]. DNA vaccines encoding TPI vaccines administered with adjuvants such as Heat shock protein 70 (HSP-70) and IL-12 have also yielded promising efficacy results, reducing worm burden and egg production by 55% and 57%, respectively [70,111].

To enhance the efficacy of the DNA-TPI vaccine, various combinations of recombinant adenoviral vectored TPI or DNA-TPI primed with IL-12 and recombinant TPI boosting strategies were conducted in mice [144,145]. The strategy was also utilized in water buffalo vaccination trials where DNA-SjTPI and DNA-Sj23 were administered with IL-12 priming and recombinant protein boost, resulting in satisfactory protective efficacy (reducing worm and egg numbers by 55% and 53%, respectively) [111]. Importantly, this DNA SjTPI-IL12 priming and rSjTPI boost vaccine approach was recently concluded in a five-year phase-3 clustered randomized control trial to determine the effect of combining vaccination of water buffaloes alongside human mass chemotherapy and snail control on the incidence of human schistosomiasis. Bovines were administered a DNA SjTPI and IL-12 vaccine intramuscularly and then boosted with the rSjTPI six months later. PZQ mass treatment in people and snail mollusciding were conducted simultaneously. There was a 16% reduction in human schistosomiasis infection detected when vaccination was combined with human treatment and a 31% reduction observed when combined with snail mollusciding, with infection detected using the Kato-Katz technique when at least one egg was observed at two follow-up visits in bovine-vaccinated villages compared with the bovine-unvaccinated villages in the Philippines [87].

### 5.4. S. japonicum 23-kDa Integral Membrane Protein (Sj23)

*S. japonicum* 23-kDa integral membrane protein (Sj23), a tetraspanin protein family member, is found in the membrane of all life cycle stages of *S. japonicum* and was noted to be particularly abundant on the tegument of the schistosomula [146,147,148,149,150]. Sj23 acts as a structural protein and may play a role in maintaining parasite membrane integrity, signal transduction, cell activation, proliferation, motility, and metastasis [43,150,151]. A high anti-Sj23 antibody was detected in 90% of schistosomiasis-infected individuals, suggesting its significant role in modulating the host immune response as a potential vaccine target [150,152].

Vaccinating mice with a combination of plasmid DNA vaccines comprising four membrane proteins, including Sj23, Sj62, Sj28, and Sj14-3-3, resulted in up to a 45% decrease in worm burden [153]. The vaccine efficacy of Sj23 was further assessed in water buffaloes and improved by administering HSP70 as an adjuvant, resulting in a significant reduction in worm burden (45–51%) and faecal egg hatching ability (47–52%) [70]. In another field trial, vaccination of water buffalo with DNA-Sj23 reduced worm numbers and liver eggs by 38% and 50%, respectively [103]. Meanwhile, administering pigs with IL-12 in conjunction with DNA-Sj23 generated increased protection against *S. japonicum* cercariae challenges, resulting in a 59% and 56% decrease in worm numbers and hepatic eggs, respectively [154]. The protective role of DNA-Sj23 was further evidenced when combined with DNA-SjFABP as a bivalent vaccine, generating a 40% reduction in worm burden (Sj23 alone) and 52% when combined with SjFABP in mice. It also generated a 38% and 61% reduction in liver eggs alone and combined with SjFABP, respectively [155]. Da’dara in 2019, immunizing water buffalo using a combined heterologous DNA prime-protein boost regimen, provided up to 55% and 53% reductions in adult worms and liver eggs, respectively, with vaccine-specific antibody responses [111].

### 5.5. S. japonicum Glutathione-S-Transferase (SjGST)

*S. japonicum* Glutathione-S-transferase (SjGST). Glutathione-S-transferases are found in plants, animals, and some helminths, including *Facsiola hepatica*, *Onchocerca volvulus*, *Necator americanus*, and the schistosomes [156]. GSTs primarily act as binding proteins, catalyzing the conjugation of glutathione to electrophilic compounds/oxidative stress ligands, thereby providing protection against immune attacks, oxidative stress, and inflammatory diseases [157,158,159]. SjGST was widely distributed in the egg, teguments, and parenchyma of the male and female adult worms, as well as eggshells [40,160]. The 26 and 28 kDa GST of *S. japonicum* have been well characterized, and the 26 kDa GST was identified as the major detoxification enzyme [157,161]. The effective anti-fecundity characteristics of the 28 kDa GST enzymes of *S. mansoni* and *S. haematobium* have been demonstrated even though the outcome of the phase three trial of Bilhvax (the ShGST vaccine) was beneath the expected efficacy benchmark [21].

In animal models, including mice, sheep, pigs, and water buffalo, the recombinant Sj26GST has proven to be an effective vaccine candidate [103,119,162,163,164,165]. SjGSTs have been administered in different routes in various research studies, orally as a bivalent vaccine with Sj23 in a *Salmonella typhirum* delivery system, resulting in a decrease in the worm (21–51%) and egg burden (32–62%) [166]. Nanoparticle-coated Sj26GST DNA vaccines have also shown promising results in murine studies, resulting in a 23–71% reduction in tissue egg burden [167]. Similarly, vaccination using a combination of DNA Sj26GST, IL-12-expressing plasmid, and recombinant Sj26GST (rSj26GST) resulted in a reduction in worm burden (62–74%), liver egg burden (17–46%), and size of tissue granuloma with high expression of both Th1 and Th2 cytokines [110]. Administration of Sj26GSTs in combination with anti-CD25 mAb, an antibody that interferes with Tregs, thereby enhancing vaccine efficacy, in mice resulted in an enhanced reduction in worm burden from 24–47% [62,65,160].

## 6. Prospective *Schistosoma japonicum* Vaccine Candidates

While murine studies have provided more understanding of the immunological reactions against schistosomiasis and vaccine effectiveness against cercarial challenge [4,47], differences still exist between the immunopathology in mice and the natural host of the parasite [124,168,169]. However, they are still necessary and normally used as the first step in evaluating the protective efficacy of vaccines in vivo due to the low cost of maintaining them and reagent availability [168]. Notwithstanding, an accurate assessment of vaccine efficacy in the natural host of the parasite is crucial to promptly deploy vaccines in affected regions and to achieve the WHO target of schistosomiasis elimination. It is, therefore, vital to secure funding for the acquisition of large animals (bovines) for vaccine challenge studies based on the results obtained from small animal trials and evaluate the efficacy of various candidates developed in the field [16]. Here, we describe some potential vaccine candidates as summarized in Table 2, which were only tested in the mouse model with the potential to further assess their protective efficacy in the natural host of the parasite (including sheep, goats, pigs, and bovines).

### 6.1. S. japonicum Insulin Receptors

Our previous studies have demonstrated that adult schistosomes are critically reliant on the insulin signalling pathway for a range of biological processes, including growth, maturation, and egg production [107,143,170,171,172,173]. Accordingly, we isolated two types of insulin receptors from *S. japonicum* (SjIR1 and SjIR2) [143], which play a critical role in regulating glucose transport from host blood into schistosomes. The binding between SjIR1 and host insulin or *S. japonicum* insulin-like peptide (SjILP) on the worm tegument [174] can activate downstream signal transduction, resulting in the import of glucose from host blood into the parasites. The glucose consumed by the worms is then transported to various cells and tissues within the parasites. Subsequently, the binding of SjILP to SjIR-2 occurs in the parenchyma of male and female vitelline cells, regulating parasite growth, pairing, and adult fertility [107,170,172]. The ligand domains of SjIR1 (SjLD1) and 2 (SjLD2) have high immunological cross-reactivity [173], sharing common functional properties. We have shown that blocking host insulin/SjILP binding to SjLDs resulted in a reduction in worm glucose uptake, causing worm starvation, growth retardation, and a subsequent decline in egg output [107]. These findings were supported by mouse vaccine/challenge trials showing that immunization of mice with SjLD2 fusion proteins induced ~42% reduction in worm length and 56–67% faecal egg reduction [175] against challenge infection. Despite there being no observable reduction in worm burden, the considerable reduction in egg production in mice vaccinated with rSjLD2 highlighted its potential to block parasite transmission. It is important to emphasize that schistosome eggs are the direct cause of pathology in host tissues and are responsible for disease transmission. Therefore, targeting fecundity and/or egg viability is a realistic strategy for developing a veterinary transmission-blocking vaccine against *S. japonicum* [25]. 

### 6.2. S. japonicum Calpains (Sj-p80)

Intracellular calcium-dependent cysteine proteases, found in almost all eukaryotes, are proteolytic proteins with two subunits [21,101,176], localized in the penetration glands of cercariae, the surface syncytial epithelium (of adult male worms), and the underlying musculature of different schistosome life stages [176,177]. Larval acetabular gland secretions contain calpain involved in the degradation of fibronectin, inhibiting blood clotting around the worm and enhancing larval migration [131,178,179,180,181]. It also promotes the generation of eggs in mature worm pairs by inducing the secretion of miRNAs in extracellular vesicles and the consumption of RBCs [178]. Vaccination with recombinant calpain induced a Th1-biased protective immune response by upregulating Interferon-gamma and inducible nitric oxide, as well as a significant reduction in worm burden (41%) [153,182]. Not surprisingly, the cross-reactivity of *S. mansoni* calpain (Smp80) was tested against *S. japonicum* infection in mice, resulting in a 47% reduction in worm burden [183,184]. Subsequently, a pilot study demonstrated the practicality of purifying recombinant *S. japonicum* calpain (Sj-p80) as a veterinary vaccine, resulting in protection against infection in mice with robust antibody titres and a reduction of eggs deposited in the liver (up to 51%) [105].

**Table 2 ijms-25-01707-t002:** Report on the protective efficacy of potential *S. japonicum* vaccine candidates using mouse models since 2012 (with >40% reduction in worm burden and >50% reduction in fecal egg output).

Candidate	Function	Immunization Strategy (Adjuvant)	Worm Burden Reduction (%)	Faecal Egg Reduction (%)	Liver Egg Reduction (%)	Intestinal Eggs Reduction (%)	Other Effects	Ref.
*S. japonicum* Calpain	Haemoglobin lysis, Inhibition of blood clotting, immune evasion.	rSm-p80 (GLA-se)	46.75		4.74	39.9	Balanced Th1 and Th2 responses.	[183]
		rSjp80 (Montanide ISA61)	31.3–46.1		51.4		29.1–31.60% decrease in miracidial hatching.	[105]
Ligand domain 1 and 2 of *S. japonicum* Insulin receptors	Glucose uptake.	rSjLD1 (Quil A)	43.5–44	47–61	ns, 44	ns, 46	10–19% reduction in worm length, 56% reduction in intestinal granuloma	[104,107]
		rSjLD1 (Montanide ISA 720)	30	68	56	48		[104]
		rSjLD2 (Quil A)	-	56–67	28 (ns)	23	16–42% reduction in mean adult worm length	[175]
*S. japonicum* Triose Phosphate isomerase	Glucose metabolism.	rSjTPI (Montanide ISA 720)	ns	51 (ns)	21 (ns)	ns		[104]
		rAdV-SjTPI.opt	36–50		41–52		Th1 and Th2 cytokines induced, IgG levels elevated, Reduced liver granuloma	[144,145,185]
		rAdV-SjTPI.opt + rSjTPI boost	72		72			
		rAdV-SjTPI.opt + rAdV-SjTPI.opt boost	44		57			
		pcDNA-SjTPI.opt + rAdV-SjTPI.opt	45.13		54.57			
*S. japonicum* membrane proteins	Regulation of cell-signalling, proliferation, adhesion, spreading, migration, and fusion.	pcDNA-Sj2-29	53.2		51.4		Increased IgG, IL-4, and IFN-γ and reduction of the area of the granuloma	[186]
*S. japonicum* Glutathione-S-transferase	Detoxification of oxidative stress ligands.	pcDNA/Sj26GST	62.02		17.08 (ns)		Increased titre of anti SjGST IgG. Induction of IL-2, IFN-γ, IL-4, and IL-10	[110]
		pcDNA/Sj26GST (IL-12)	68.99		37.54		Increased titre of anti SjGST IgG. Induction of IL-2, IFN-γ, IL-4, and IL-10 Increased IFN-γ, IL-2	[110]
		pcDNA/Sj26GST + rSjGST	72.33		34.46			[110]
		rSj26GST (anti-CD25 mAb)	47.09		50			[65]
*S. japonicum* Fatty acid binding protein	Binds and transports long-chain fatty acids, Larval migration, and immune evasion.	rSjFABP (Freund, anti-CD25 mAb)	56.08		53.29		Increased IFN-γ, IL-2, Il-4, and Il-5	[63]
		SjFABP (Freund’s, anti-CTLA-4 mAb)	54.61		59.25			[66]
*S. japonicum* glyceraldehyde-3-phosphate dehydrogenase	Energy metabolism.	rSjGAPDH (anti-CTLA-4 mAb)	59.36		62.45		Increased IFN-γ, IL-2, Il-4, and Il-5	[64]
*S. japonicum* Adenylate kinase 1	Energy generation.	rSjAK1 (Freund)	50		40		56% reduction in granuloma area. Increased IFN-γ and IL-2.	[187]
*S. japonicum* Lethal Giant Larvae	Apical-basal cell polarity, cell proliferation, differentiation, and tissue organization.	rSjLGL (ISA206)	12.91–50.0		29.23–49.86		53–75% reduction in egg hatching rate. High IgG titre.	[188]
*S. japonicum* inhibitor apoptosis protein (SjIAP)	Regulates cell apoptosis.	Ad-rSjIAP	34.4–41.5		23.8–39.6%		Increased IFN-γ, IL-2, IL-4, and IL-10	[189]

Note: AdV—adenoviral vectored; LD—Ligand domain; mAb—monoclonal antibody; ns—Not significant; pcDNA—plasmid DNA; IL—Interleukins; r—Recombinant; IFN—Interferon; Ig—Immunoglobulin; Sj—*S. japonicum*; Sm—*S. mansoni*.

### 6.3. S. japonicum Glyceraldehyde Dehydrogenase (SjGAPDH)

*S. japonicum* Glyceraldehyde dehydrogenase (SjGAPDH) is a glycolytic enzyme important for the energy provision needed for the growth of the schistosomula [190]. It is expressed during all life cycle stages of the parasite, especially the schistosomula, and plays a critical role in the modification of intracellular signalling, activation and binding to plasminogen, inhibition of coagulation, and facilitation of the schistosome invasion and migration in the host [64,190,191,192]. A combination of SjGAPDH with adjuvants like anti-CD25mAb and anti-CTLA-4 monoclonal antibodies resulted in sufficient protective efficacy against challenge infection [64,112,193]. Tang showed that administering anti-CD25 mAb enhanced the protective efficacy of rSjGAPDH by reducing worm and egg burden from 32 to 60% and 35% to 59%, respectively [112]. Additionally, the worm reduction rate increased from 27% to 55% when anti-CTLA-4 was administered with rSjGAPDH with the induction of the Th1 immune response [64].

### 6.4. S. japonicum Fatty Acid Binding Proteins (Sj14/SjFABP)

*S. japonicum* Fatty acid binding proteins (Sj14/SjFABP) are 18-kDa cell membrane proteins that bind long chain fatty acids and sterols, enabling their extraction from the host and usage for cell signalling, metabolic pathway, and inflammatory pathways [63,194,195]. SjFABP is localized in the parenchyma, digestive tract of adult worms, and eggshells, indicating its critical role in egg development and maturation [195]. A previous study showed that mice injected with a bivalent DNA vaccine combining SjFABP and Sj23 can generate a 52% and 61% reduction in worm number and liver egg burden, respectively, while a 42% and 39% reduction in worm and liver egg burden were observed in mice vaccinated by SjFABP alone [155]. Co-administration of the monoclonal anti-CD25 mAb or anti-CTLA-4 mAb with recombinant SjFABP increased worm reduction from 30% to 56% and 54%, respectively, and reduced egg burden from 39% to 53% and 59%, respectively [63,66].

## 7. Challenges and Future Trends in the Development of Anti-Schistosome Vaccines

The development of transmission-blocking vaccines has been stifled by several challenges, including the complex life cycle of the parasite and its ability to evade the host immune response [124,131]. Another challenge is the need to identify a vaccine target that can induce both Th1 and Th2 responses, along with selecting the appropriate adjuvants and delivery systems to enhance the immune response [95]. However, recent advancements, through ‘omics’ technologies, have identified potential vaccine targets [196]. Additionally, relying solely on vaccine trials in mouse models may not provide the most reliable data, as some parasites do not reach maturity in portal veins and are terminated in the mice’s lungs from natural causes such as fragile pulmonary capillaries and parasite demise in alveoli [169]. Also, the vaccine experiment usually involves a single challenge infection in contrast to repeated exposures that may occur in nature. Unfortunately, funding for schistosomiasis, a neglected tropical disease [18], is limited, and funding for high-technology solutions, such as vaccine development for deployment in reservoir hosts, remains limited. Moreover, it is important to consider the potential risks associated with the induction of inappropriate immune responses through the induction of IgE antibodies and mast cell responses in vaccine development. While successful vaccine trials indicate that IgE is necessary for protection, the risk of anaphylaxis is something that needs to be taken into account [124]. Furthermore, the large-scale production of recombinant proteins is a challenging factor [18,139]. Nonetheless, the introduction of mRNA vaccines allows for large-scale production of purified vaccines relatively quickly and with high yields [197]. This advancement has been particularly helpful in the fight against COVID-19, where the rapid development and production of vaccines have been essential in controlling the spread of the virus.

A revolutionary anti-schistosome vaccine production strategy is required to effectively determine and test novel vaccine targets at high throughput. To further characterise novel vaccine targets, understand the host-parasite relationship, cellular interactions, and mechanism of growth and development of schistosomes, cutting-edge technologies such as transcriptomics, proteomics, exosomics, immunomics, RNA interference, and CRIPSR-Cas (Clustered regularly interspaced short palindromic repeats-CRISPR associated) mediated approaches have been employed to achieve this [40,170,196,198,199,200,201,202,203,204,205,206,207,208,209,210,211].

Proteomics has been widely applied in identifying life cycle stage-specific proteins as vaccine targets and potential diagnostic biomarkers [207,209,212,213]. Similarly, RNA sequencing analysis has been utilised to understand the immune cell response to infection, re-infection, and vaccine interaction through cellular transcriptome profiling. CRISPR-Cas, a gene-editing tool, provides a more effective, efficient, and accurate method of genome editing with potential application in Schistosoma research by adding to the available tool for discovering new vaccine targets and predicting host response to vaccines [201,214,215,216]. The recently completed genomic sequences of schistosomes [217,218,219] provide a solid study base for the identification of more potential schistosome genes/proteins as vaccine targets.

Following the unparalleled success of the SARS-CoV-2 mRNA vaccines, the exploration of mRNA vaccines emerges as a solution to the prevention and elimination of parasitic diseases. The advantages of mRNA compared to other conventional vaccines include the simple and fast development process with high yield, the synthesis of natural antigens in vivo, and the generation of robust B- and T-cell immune responses [197,220,221]. mRNA vaccines can also be applied as multivalent vaccines protecting more than one infectious agent and can induce both Th1 and Th2 cell-mediated immune responses [220,222,223]. The vaccine formulations are also safer without risk of integration into the host’s genome and are easier to deliver to the cytosol than DNA vaccines, which also need to cross nuclear membranes [223,224,225]. In vivo-translated proteins have the same natural conformation and glycosylation as those from the live pathogen, thus favouring the production of highly specific antibodies [226]. With rapid advancements in mRNA in vitro transcription (IVT) technologies, challenges posed by using mRNA vaccines, such as their size and charge (affecting their uptake into cells) and susceptibility to nucleases and immune cells, have been considerably resolved with the use of delivery carriers such as lipid nanoparticles (LNP), the modification of some nucleosides chemically with analogues such as pseudouridine, and the use of optimized codons [197,224,227].

To date, mRNA-based anti-protozoan parasite vaccines, including different species of *Plasmodium* malaria [228,229], *Leishmania donovani* [230], and *Toxoplasma gondii* [231], have been reported, but none as yet against any helminth infection. Promisingly, a mRNA multivalent vaccine against *Ixodes scapularis*, the black-legged tick, has recently been developed [232], which produced highly specific antibodies and stimulated robust protection against tick challenge [232], highlighting the potential for developing a multivalent mRNA vaccine against the multicellular organism. Recent research on multivalent COVID mRNA vaccines targeting multiple spike mutations of coronavirus [233] further provides a way forward to overcome the obstacle that has hindered the development of anti-schistosome vaccines [234]. This multivalent strategy has the potential to stimulate synergistic effects and further enhance protective immune responses [197], providing the opportunity to develop veterinary transmission-blocking multivalent vaccines against several parasite infections and other major zoonotic diseases transmitted between animals and humans. It has the prospect of leading to a new approach to the control of zoonotic infections and the achievement of the One Health goal.

## 8. Conclusions

Developing transmission-blocking vaccines against zoonotic schistosomiasis is critical for improved control, resulting in an effective and feasible public health outcome that aligns with the One Health concept to attain optimal health for people, animals, and the environment. This strategy will assist in the long-term prevention of human and animal infections. Taking advantage of current technological advancements and deploying some potential vaccine candidates in the field to be trialled in water buffalo, the natural host, is a critical step in the direction of achieving this goal. The incorporation of a veterinary-based transmission vaccine, coupled with complementary interventions such as MDA, improved sanitation and hygiene, health education, and snail control, would be invaluable to achieving the elimination of zoonotic schistosomiasis.

## Figures and Tables

**Figure 1 ijms-25-01707-f001:**
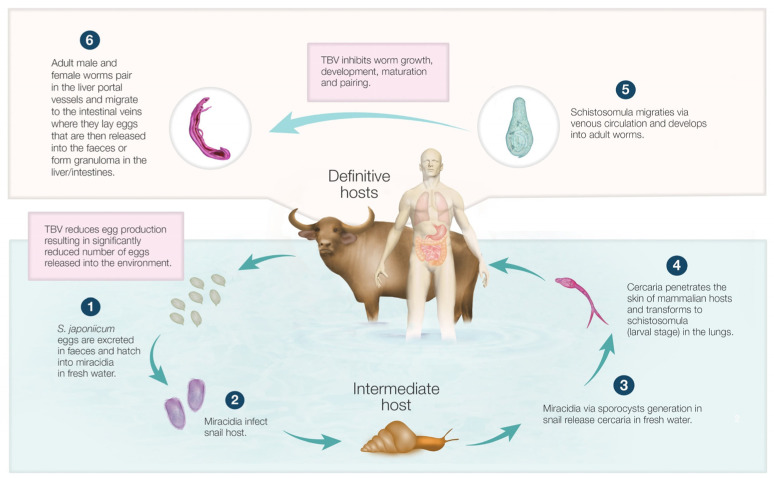
Life cycle of *Schistosoma japonicum*. Eggs released into the environment hatch, with each releasing a miracidium into fresh water. The miracidium penetrates the *Oncomelania* snail. Sporocysts develop in infected snail hosts and release cercariae in the water. Cercariae penetrate the skin of humans or animals upon contact with contaminated water, migrating through blood vessels to the lungs and then the liver, where male and female worms pair. Mature worms move to the mesenteric veins of the intestines and lay eggs, which are then shed in faeces or deposited in the liver or intestine, causing granuloma disease in tissues. Transmission-blocking vaccines (TBV) are proposed to reduce environmental egg contamination by inhibiting worm growth, development, maturation, and egg production in mammalian hosts (especially water buffalo, which are responsible for up to 90% of the faecal egg environmental contamination), thereby significantly reducing the number of faecal egg outputs into the environment. See Section 3 on the rationale for developing TBVs.

**Table 1 ijms-25-01707-t001:** Veterinary transmission-blocking vaccine candidates in water buffalo (with >45% reduction in worm burden/faecal egg output) against *Schistosoma japonicum* challenge.

Antigen	Immunization Strategy	Method of Immunization Regimen	Worm Burden Reduction %	Faecal Egg (FER) or Miracidia Hatching (MH) Reduction %	Ref.
Paramyosin	Recombinant	Montanide, ISA 206	52–61		[118]
SjTPI	DNA	Fused to bovine HSP70	41–51	33–52 (MH)	[70]
Sj23	DNA	HSP70 and boosted with a plasmid DNA encoding IL-12	45–51	47–52 (MH)	[70]
Sj23	DNA		38	50 (MH), 60 (FE)	[103]
Combination of SjTPI-Sj23	DNA	Administered with pIL-12 and boosted with a combination of rSjC23 and rSjTPI	55	57 (MH)	[111]
Combination of Sj23-Hsp70	DNA	Administered with a plasmid DNA encoding IL-12	37–41	31–46 (MH)	[111]
Sj26GST	Recombinant protein		22	50 (FE)	[119]
Sj28GST	Recombinant protein	Injected with Freund’s adjuvants	37	62 (MH)	[103]
Sj28	DNA		39	62(MH)	[103]
Cryopreserved-irradiated/freeze-thaw schistosomula	Whole	Single/twice intradermal vaccination	62–65		[120]
Cercariae UV irradiated	Whole	Six immunizations	89		[121]

Note: SjTPI, *S. japonicum* Triose phosphate Isomerase; Sj23, *S. japonicum* 23-kDa integral membrane protein; Sj28GST, *S. japonicum* 28-kDa subunit glutathione S-transferase; Sj26GST, *S. japonicum* 26-kDa subunit glutathione S-transferase; HSP70, bovine heat shock protein 70.

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
