# Peer review of "Transmission-Blocking Vaccines against Schistosomiasis Japonica"

_ijms, 2024, doi:10.3390/ijms25031707_

Round 1

Reviewer 1 Report

Comments and Suggestions for Authors

The work is a valuable contribution to the field, offering a well-organized and scientifically sound overview of the challenges associated with schistosomiasis japonica and the potential of veterinary transmission-blocking vaccines as part of integrated control strategies.

While the document is comprehensive, some sections could benefit from additional clarity or further elaboration. For instance, the discussion on adjuvants could be expanded to provide more details on specific adjuvants and their mechanisms of action. Additionally, the document briefly touches on the current status of transmission-blocking vaccines but could provide more in-depth information on recent advancements, challenges, and future directions in this area.

Comments on the Quality of English Language

Minor editing of English language required

Author Response

Reviewer #1: The work is a valuable contribution to the field, offering a well-organized and scientifically sound overview of the challenges associated with schistosomiasis japonica and the potential of veterinary transmission-blocking vaccines as part of integrated control strategies.

1. While the document is comprehensive, some sections could benefit from additional clarity or further elaboration. For instance, the discussion on adjuvants could be expanded to provide more details on specific adjuvants and their mechanisms of action. Additionally, the document briefly touches on the current status of transmission-blocking vaccines but could provide more in-depth information on recent advancements, challenges, and future directions in this area.

Response: Thank you for this valuable insight. More details on specific adjuvants and their mechanisms of action have been elaborated in lines 230-301.

 In addition, more information on recent advancements, challenges, and future directions have been discussed and added to subheading 7 “ Challenges and future trends in the development of anti-schistosome vaccine….” -  lines 576-598.

2. While the document is comprehensive, some sections could benefit from additional clarity or further elaboration. For instance, the discussion on adjuvants could be expanded to provide more details on specific adjuvants and their mechanisms of action. Additionally, the document briefly touches on the current status of transmission-blocking vaccines but could provide more in-depth information on recent advancements, challenges, and future directions in this area.

Response: Thank you for this valuable insight. More details on specific adjuvants and their mechanisms of action have been elaborated in lines 230-301.

 In addition, more information on recent advancements, challenges, and future directions have been discussed and added to subheading 7 “ Challenges and future trends in the development of anti-schistosome vaccine….” -  lines 576-598.

Reviewer 2 Report

Comments and Suggestions for Authors

Dear authors,

Thank you for this notable research work. Below are the comments and suggestions on the research paper entitled “Transmission-blocking Vaccines against Schistosomiasis japonica”. 

The manuscript is well-written and has presented important information on the need to develop a transmission-blocking vaccine for bovines to control zoonotic schistosomiasis effectively.

However, the following comments must be addressed to further improve the research paper.

Line 2: The scientific name Schistosomiasis japonica must be written in one word. 

Line 108-109: Kindly enhance the resolution of Figure 1 Life cycle of Schistosoma japonicum

Line 417-418: The title or description of Table 2 must be near the figure. 

It would be good to highlight in the paper the implications and applicability of anti-schistosome vaccines in endemic areas. How will the endemic community benefit from this? 

Discuss further the role of bovines in the zoonotic transmission of schistosomiasis. Bovines exhibit a self-cure phenomenon for Schistosoma japonicum infection. What is the potential impact of this in the control and elimination of zoonotic schistosomiasis? 

These are some of the suggestions to further improve the manuscript and substantiate the relevance of the research paper. Kindly highlight in red the revisions made in the revised manuscript for confirmation upon submission of the revised paper.

Thank you for your research work.    

Author Response

Reviewer #2: Thank you for this notable research work. Below are the comments and suggestions on the research paper entitled “Transmission-blocking Vaccines against Schistosomiasis japonica”.

The manuscript is well written and has presented important information on the need to develop a transmission-blocking vaccine for bovines to control zoonotic schistosomiasis effectively.

However, the following comments must be addressed to further improve the research paper

1. Line 2: The scientific name Schistosomiasis japonica must be written in one word.

Response: Apologies for that. It has been modified now.

2. Line 108-109: Kindly enhance the resolution of Figure 1 Life cycle of Schistosoma japonicum.

Response: Thank you very much for the suggestion. The resolution of Figure 1 has been adjusted to 300dpi as attached.

3. Line 417-418: The title or description of Table 2 must be near the figure.

Response: Thanks for the comments. Modifications have been made to Table 2.

4. It would be good to highlight in the paper the implications and applicability of anti-schistosome vaccines in endemic areas. How will the endemic community benefit from this?

Response: Thanks for the suggestions. The implications and the endemic community benefit have been highlighted in lines 188-197.

5. Discuss further the role of bovines in the zoonotic transmission of schistosomiasis. Bovines exhibit a self-cure phenomenon for Schistosoma japonicum infection. What is the potential impact of this in the control and elimination of zoonotic schistosomiasis?

Response: Thank you for the feedback. As requested, we have discussed more about the important role of bovines in the transmission and its impact in the control of zoonotic schistosomiasis in lines 169-197. The self-cure phenomenon observed in older bovines has also been discussed in lines 146-152 

6. These are some of the suggestions to further improve the manuscript and substantiate the relevance of the research paper. Kindly highlight in red the revisions made in the revised manuscript for confirmation upon submission of the revised paper.

Response: Thank you for your valuable feedback. The revisions have been highlighted in red.

Reviewer 3 Report

Comments and Suggestions for Authors

1)  This review article will be contributed for one of the schistosome biological aspects, and it may be helpful for a development of the countermeasures of schitomeasis.

2)  The authors descrived something about veterinary field, and they gave its detailed epidemiological data on only water buffaloes .   However, the field is wider, so they had better add the other animals, eg., dogs and rats, as well.

3)  For example, according to Matsumoto et al. (1999): , Jpn. J. Trop. Med. Hyg., Vol. 27, No. 2, 1999, pp. 175-180 175, "stool examination or necropsy of animal reservoirs in San Pedro, San Narciso and Malabo (of Philippines) were as follows; dogs: 9.7%, 7.4% and 19.2%; rats: 10.4%, 8.7% and 26.1%, respectively. Water buffaloes were all negative", they may collect the other papers.

Author Response

Reviewer #3: This review article will be contributed for one of the schistosome biological aspects, and it may be helpful for a development of the countermeasures of schistomiasis.

1. The authors described something about veterinary field, and they gave its detailed epidemiological data on only water buffaloes. However, the field is wider, so they had better add the other animals, eg., dogs and rats, as well. 

Response: Thank you for the comments.  This has been addressed in lines 176-184.

2. For example, according to Matsumoto et al. (1999): , Jpn. J. Trop. Med. Hyg., Vol. 27, No. 2, 1999, pp. 175-180 175, "stool examination or necropsy of animal reservoirs in San Pedro, San Narciso and Malabo (of Philippines) were as follows; dogs: 9.7%, 7.4% and 19.2%; rats: 10.4%, 8.7% and 26.1%, respectively. Water buffaloes were all negative", they may collect the other papers.

Response: Thank you for this recommendation. I have cited this paper as well as other relevant papers in lines 176-184.